# Understanding and countering the spread of conspiracy theories in social networks: Evidence from epidemiological models of Twitter data

**Julian Kauk**[1]*, **Helene Kreysa**[1], **Stefan R. Schweinberger**[1,2]

**1** Department of General Psychology and Cognitive Neuroscience, Friedrich Schiller University Jena, Jena, Germany, **2** DFG Research Unit Person Perception, Friedrich Schiller University Jena, Jena, Germany

* julian.kauk@uni-jena.de

## Abstract

Conspiracy theories in social networks are considered to have adverse effects on individuals' compliance with public health measures in the context of a pandemic situation. A deeper understanding of how conspiracy theories propagate through social networks is critical for the development of countermeasures. The present work focuses on a novel approach to characterize the propagation of conspiracy theories through social networks by applying epidemiological models to Twitter data. A Twitter dataset was searched for tweets containing hashtags indicating belief in the "5GCoronavirus" conspiracy theory, which states that the COVID-19 pandemic is a result of, or enhanced by, the enrollment of the 5G mobile network. Despite the absence of any scientific evidence, the "5GCoronavirus" conspiracy theory propagated rapidly through Twitter, beginning at the end of January, followed by a peak at the beginning of April, and ceasing/disappearing approximately at the end of June 2020. An epidemic SIR (Susceptible-Infected-Removed) model was fitted to this time series with acceptable model fit, indicating parallels between the propagation of conspiracy theories in social networks and infectious diseases. Extended SIR models were used to simulate the effects that two specific countermeasures, fact-checking and tweet-deletion, could have had on the propagation of the conspiracy theory. Our simulations indicate that fact-checking is an effective mechanism in an early stage of conspiracy theory diffusion, while tweet-deletion shows only moderate efficacy but is less time-sensitive. More generally, an early response is critical to gain control over the spread of conspiracy theories through social networks. We conclude that an early response combined with strong fact-checking and a moderate level of deletion of problematic posts is a promising strategy to fight conspiracy theories in social networks. Results are discussed with respect to their theoretical validity and generalizability.

**Data Availability Statement:** The data underlying the results presented in the study are available from (https://osf.io/rha3w/). The data stored in the .rds file is compliant with Twitter's policy

information, as it contains only frequencies of hashtags but no tweet data itself.

**Funding:** The author(s) received no specific funding for this work.

**Competing interests:** The authors have declared that no competing interests exist.

## Introduction

The COVID-19 pandemic has been accompanied by an emerging stream of misinformation in social networks [1]. The World Health Organization (WHO) explicitly noted the need to manage the "infodemic", i.e. to avoid a state of overabundance of information [2]. While (true) news play an important role in informing the public, misinformation can undermine the public health responses and can therefore significantly affect adherence to hygiene recommendations and efficacy of countermeasures [3]. The effects of misinformation on pandemic-related outcome measures like incidence or mortality remain to be estimated, but it is reasonable to assume an adverse impact on both the spread of severe acute respiratory syndrome coronavirus 2 (SARS-CoV-2) [4] and efficient public health countermeasures [5]. When targeting misinformation, it is essential to consider social networks, as they have been shown to be important amplifiers [6]. There is an emerging body of research about the diffusion and prevalence of misinformation within social networks [7]. With respect to Twitter, Bovet and Maske [8] demonstrated for data from the 2016 US presidential election that the percentage of tweets containing misinformation can be up to 25%. Similar fake news on this occasion were also spread via other social media such as Facebook [9]. Shin et al. [10] investigated the temporal dynamics of rumors on Twitter, revealing that false political rumors seem to reappear, whereas true rumors disappear after a short time period. More precisely, false political rumors had an average of 3.31 peaks whereas true rumors seem to appear only once. Consequently, true rumors showed significant "burstiness", meaning that nearly half of the total tweet volume (49.58% on average) was observed on a single day. There is also evidence that "echo chambers", i.e. the formation of groups where a shared belief is framed and reinforced [11], play a significant role in the amplification of misinformation [12]. Even though social media platforms have put effort into updating their algorithms in order to limit the spread of misinformation, misinformation remains a constant source of problems: negative effects on adherence, democracy and diversification can be expected [13], which are potentially pervasive and long-lasting.

A better understanding of *how* misinformation propagates through social networks is a critical ingredient, since improper countermeasures may fail to work effectively. The idea that the spread of rumors can be modeled within an epidemiological framework emerged for the first time in a comment in 1964 [14], pointing out that infectiological states within an epidemiological SIR model (S: Susceptible, I: Infected and R: Removed) can be reinterpreted to fit for rumors. More recent works, for instance from Jin et al. [15], showed that more complex, adopted models may be more precise in describing rumor diffusion in social networks. Jin et al. used a SEIZ model (S: Susceptible, E: Exposed, I: Infected and Z: Skeptic; see also [16]) to characterize the spread of both (true) news and rumors on Twitter. They found that (i) their SEIZ model fitted better to the data than a simple SI model (S: Susceptible, I: Infected) and (ii) (true) news and rumors could be distinguished on the basis of the estimated model parameters. In general, previous research (see e.g. [17–20]) has focused on model identification, aiming to identify epidemiological models which can characterize the diffusion of misinformation in social networks accurately.

The focus of the present study, however, is to simulate how countermeasures could affect the diffusion of misinformation in social networks. Therefore, we studied the "5GCoronavirus" conspiracy theory that emerged in January 2020 and stated that the spread of SARS-CoV-2 is caused/enhanced by the enrollment of the 5G mobile network [21, for a detailed description]. Tragically, this conspiracy theory was not restricted to social networks but apparently caused substantial physical damage, as more than 70 mobile phone masts were attacked by supporters in the United Kingdom alone [22, 23, for two newspaper reports about this

issue]. This damage to critical telecommunication infrastructure was preceded by an escalating situation on social networks, where users were called on to "break down" the 5G masts [24]. It is reasonable to assume that countermeasures targeting this conspiracy theory on social networks could have reduced or even prevented the vandalism. Accordingly, we here studied how post deletion and fact-checking could have modulated the diffusion of the conspiracy theory in social networks.

There is ample evidence that social media platforms rely on fact-checking: Facebook, for instance, explicitly describes its fact-checking strategy [25]: Qualified fact-checkers identify and review suspicious posts, and specific countermeasures, e.g. reduced distribution of problematic posts and misinformation labels, may subsequently be implemented. Recent studies indicate that fact-checking on Facebook can be moderately beneficial [26–28]. Twitter uses labels and contextual cues to address the problem of misinformation [29] and announced in January 2021 that a "community-based approach to misinformation", namely "birdwatch", will be tested from now [30]. However, whether or not fact-checking is a sufficient tool to contain the spread of misinformation in social networks remains an open question [31]. We here understand fact-checking in a broad sense, meaning that the facts disproving a conspiracy theory are presented to the users in a preventive fashion, thus resembling more general public health communication.

In addition, the deletion of posts may be a promising tool as well [32], although there is evidently an indistinct line between responsible post deletion and censorship. If post deletion is applied circumspectly and in accordance with appropriate use policies, there might be acceptance for this countermeasure within the social media communities. Recent advances in the detection of misinformation on social networks [32–35, for instance] may constitute a crucial ingredient to address the problem and can be considered as a necessary condition to perform both fact-checking and post deletion effectively. Twitter states that it performs removal of problematic tweets if these tweets evidently transport harmful contents [36].

We used advanced epidemiological models on a Twitter dataset in order to identify conditions under which countermeasures could have attenuated the spread of the conspiracy theory effectively. More precisely, we formulated a basic epidemiological model in order to characterize the diffusion of the "5GCoronavirus" conspiracy theory through Twitter. Subsequently, we incorporated both fact-checking and tweet deletion as well as a response lag into the basic epidemiological model, aiming to build up evidence about whether or not these countermeasures would have been capable of stopping the spread of the '5GCoronavirus" conspiracy theory through Twitter.

## Methods

### Basic epidemiological model

We used a simple SIR compartment model (without vital dynamics) [37] to characterize the propagation of the "5GCoronavirus" conspiracy theory. Although other authors have suggested different models to characterize the dynamics of misinformation diffusion through social networks [15, 38, for instance], we decided to use the SIR model because (i) it is less complex and deals with only five parameters, (ii) it may constitute a good starting point to incorporate countermeasures without creating complex interactions, and (iii) other researchers have used the SIR model for the same purpose as well [18, 20]. The SIR model flow can be described by

$$S \xrightarrow{\frac{\beta SI}{N}} I \xrightarrow{\alpha I} R,$$ 
(1)

**Table 1. Redefining the SIR model for conspiracy theories.**

| Scope | N | Compartments | | |
|---|---|---|---|---|
| | | S | I | R |
| Epidemiology | total population | susceptible to an infectious disease | being infected | removed due to immunity or death |
| Conspiracy Theory (in general) | total population | susceptible to a conspiracy theory | believing conspiracy theory | not believing conspiracy theory anymore |
| Conspiracy Theory (Twitter) | Theoretically: All active twitter users; *De facto*: Unclear (e.g. [15]) | have not posted a supporting tweet | have posted a supporting tweet | conspiracy theory is out of mind/ forgotten [40–42] |

where compartments $S(t)$, $I(t)$, and $R(t)$ sum up to the total population $N$, $\beta$ is the infection rate, and $\alpha$ is the recovery rate. The greater $\beta$, the more likely an infected individual infects at least one susceptible individual. The smaller $\alpha$, the longer an individual remains infectious and not recovered (infection period is given by $\frac{1}{\alpha}$). The dynamics of the model can be characterized by a set of ordinary differential equations (ODEs; [39]). The set of ODEs is given by

$$\frac{dS}{dt} = -\frac{\beta SI}{N}$$

$$\frac{dI}{dt} = \frac{\beta SI}{N} - \alpha I \quad\quad (2)$$

$$\frac{dR}{dt} = \alpha I.$$

As shown in Table 1 (top and middle row), the SIR model can be redefined fairly easily to apply to conspiracy theories. However, the strict redefinition of the R compartment is problematic, as there is no immune system "curing" false beliefs. In consequence, the R compartment needs further elaboration. It is reasonable to assume that the cognitive and communicative efforts an individual puts into a conspiracy theory both decline after a given period of time due to forgetting, habituation or decreasing interest [40–42]. We therefore assume that the conspiracy theory is out of mind after a given time period, and consequently the individual progresses to the R compartment. With respect to the propagation of conspiracy theories specifically on Twitter, a few more aspects had to be considered in the definition of the model. Susceptible individuals are considered to be susceptible to a conspiracy theory, but have not posted a corresponding tweet yet. Infected individuals are considered to believe a conspiracy theory, indicated by having posted a corresponding tweet. Table 1 (bottom row) summarizes the SIR model adapted for Twitter.

### Extended epidemiological models

In order to incorporate potential countermeasures into the basic SIR model, we identified three extended SIR models which include deletion of tweets, fact-checking and both countermeasures combined. Additionally, in order to account for delays in the beginning of countermeasures, we introduced the parameter $\delta$, which reflects the delay until countermeasures are implemented. The following sections provide a detailed formulation and description of these extended SIR models. Note that, if not indicated otherwise, the parameters (and their definitions) of the basic SIR model (described above) apply to all extended SIR models.

**SIR model with fact-checking.** The SIR model with fact-checking ($SIR_{fact-checking}$) extends the basic SIR model by the parameter $\gamma$, representing a constant ratio of individuals progressing directly from compartment S to R per time unit. The assumption of a constant ratio is

simplistic, but may hold for relatively small values of $\gamma$, as it is reasonable to assume that there is a significant proportion of susceptible individuals who are responsive to fact-checking. In order to consider the delay parameter $\delta$ in the model, we introduce the variable $\gamma_\delta$, which is a function of $t$ and $\delta$ and is "activated" only if the delay $\delta$ is exceeded. This variable is given by

$$\gamma_\delta = \begin{cases} \gamma & if \ t > \delta \\ 0 & otherwise. \end{cases} \tag{3}$$

The flow of the model therefore can be written as

$$S \xrightarrow{\frac{\beta S I}{N}} I \xrightarrow{\alpha I} R \qquad \overset{\gamma_\delta \, S}{\frown} \tag{4}$$

and the set of ODEs is given by

$$\begin{aligned} \frac{dS}{dt} &= -\frac{\beta SI}{N} - \gamma_\delta S \\ \frac{dI}{dt} &= \frac{\beta SI}{N} - \alpha I \\ \frac{dR}{dt} &= \alpha I + \gamma_\delta S. \end{aligned} \tag{5}$$

Accordingly, $\gamma$ controls how effectively susceptible individuals can be prevented from becoming convinced by a conspiracy theory.

**SIR model with tweet deletion.** The SIR model with tweet deletion ($SIR_{deletion}$) is similar to the basic SIR model except for an additional parameter $\zeta$, representing a constant ratio of individuals being moved from compartment $I$ to compartment $R$ per time unit. The assumption of a constant ratio is simplistic as well, but may resemble the limited capabilities of a social media platform to detect and delete suspect tweets. The definition of the variable $\zeta_\delta$ is equivalent to $\gamma_\delta$ except that $\gamma$ is replaced by $\zeta$. The variable is therefore given by

$$\zeta_\delta = \begin{cases} \zeta & if \ t > \delta \\ 0 & otherwise. \end{cases} \tag{6}$$

The flow of the model is given by

$$S \xrightarrow{\frac{\beta S I}{N}} I \xrightarrow{(\alpha + \zeta_\delta)I} R \tag{7}$$

and the set of ODEs can be written as

$$\begin{aligned} \frac{dS}{dt} &= -\frac{\beta SI}{N} \\ \frac{dI}{dt} &= \frac{\beta SI}{N} - (\alpha + \zeta_\delta)I \\ \frac{dR}{dt} &= (\alpha + \zeta_\delta)I. \end{aligned} \tag{8}$$

Parameter $\zeta$ therefore controls how many individuals who are convinced by a conspiracy theory move to the $R$ compartment by deletion of their tweets.

**SIR model with mixed countermeasures.** SIR model with mixed countermeasures ($SIR_{mixed}$) combines $SIR_{fact-checking}$ and $SIR_{deletion}$ by introducing both $\gamma$ and $\zeta$ to the model. Note that the definitions of both parameters (and their corresponding variables $\gamma_\delta$ and $\zeta_\delta$) also apply to the $SIR_{mixed}$ model. The flow can be described by

$$S \xrightarrow[\frac{\beta S I}{N}]{\overbrace{\hspace{3cm}}^{\gamma_\delta S}} I \xrightarrow{(\alpha+\zeta_\delta)I} R \tag{9}$$

and the set of ODEs is given by

$$\begin{aligned} \frac{dS}{dt} &= -\frac{\beta SI}{N} - \gamma_\delta S \\ \frac{dI}{dt} &= \frac{\beta SI}{N} - (\alpha + \zeta_\delta)I \\ \frac{dR}{dt} &= (\alpha + \zeta_\delta)I + \gamma_\delta S. \end{aligned} \tag{10}$$

## Parameter identification

The following sections provide detailed information about the procedure of parameter identification for both the basic SIR model and the extended SIR models.

**Basic SIR model.** As both $\beta$ and $\alpha$ cannot be deduced properly from previous research, these parameters were treated as unknowns and had to be estimated empirically. These estimated parameters of the basic SIR model were also used subsequently to parametrize the extended SIR models. The total population size $N$ was also not specified in advance, as $N$ is theoretically given by any Twitter user who could have been exposed to the "5GCoronavirus" conspiracy theory, but *de facto* $N$ is not easy to estimate [15, 16, for a detailed reasoning]. Accordingly, the initial numbers of individuals in the compartments, $S(t_0)$, $I(t_0)$, $R(t_0)$, were treated as unknowns too.

Parameter Identification was done in $R$ [43, version 4.0.1 (2020-06-06)], using the *EpiModel* package [44] for model specification combined with the *R stats* function *optim* based on the Nelder–Mead algorithm [45] for identification of optimal parameter values. We set the initial values ("Guesses") of the parameters to be optimized arbitrarily (see S1 Table), but used as references (i) the total incidence, (ii) the assumption that only a minority of individuals is receptive to conspiracy theories at all and (iii) the assumption that the number of infected individuals is initially small. A least squared criterion was applied as given by

$$\hat{\theta} = \underset{\Theta}{\arg\min}\left(\sum_{i=0}^{t_{max}} (I_{cum}(t_i) - \hat{I}_{cum}(t_i))^2\right), \tag{11}$$

where $\hat{\theta}$ refers to the vector of estimated parameter values, $\Theta$ is the parameter space, $t_{max}$ refers to the most recent date included in the analysis, $t_i$ is the $i$-th date, $I_{cum}(t_i)$ is the cumulated incidence up to $t_i$, and $\hat{I}_{cum}(t_i)$ represents the cumulated incidence according to the model up to $t_i$.

In order to evaluate the basic SIR model, we report relative error in 2-norm as given by

$$\epsilon_{rel} = \frac{\|I_{cum}(t) - \hat{I}_{cum}(t)\|_2}{\|I_{cum}(t)\|_2}, \tag{12}$$

as well as the mean absolute error (*MAE*) described by

$$MAE = \frac{\|I_{cum}(t) - \hat{I}_{cum}(t)\|_1}{n}, \tag{13}$$

where *n* stands for the number of data points.

**Extended SIR models.** The identified parameter values of the basic SIR model were used to parametrize the extended SIR models. Consequently, the parametrized extended SIR models are similar to the parametrized basic SIR model, except for their additional parameters $\gamma$, $\delta$ and $\zeta$, where the respective parameter values were defined *a priori*. In order to identify realistic parameter values for $\gamma$, $\delta$ and $\zeta$, we had to take into account the time scale of the "5GCoronavirus" conspiracy theory, which is characterized by the onset $t_0$ at the end of January 2020 and an incidence peak $t_{IMax}$ at the beginning of April 2020 [24]. In the interest of simplicity, we here elaborated three explicit levels for each parameter, even though the parameters are, in principle, considered to be continuous.

*Countermeasures' delay $\delta$.* It is reasonable to assume different tempi when considering the response of social networks to conspiracy theories. A response may be considered as *Early*, *Delayed* or *Late*. We here define an early response as two weeks from $t_0$, while a delayed response is defined as six weeks from $t_0$. A late response is peak-adjusted and defined as two weeks before $t_{IMax}$. We therefore here studied three levels of $\delta$ as given by $\delta_1 = 14$, $\delta_2 = 42$ and $\delta_3 = t_{IMax} - 14$.

*Fact-checking parameter $\gamma$.* The parameter $\gamma$ reflects how effectively individuals can be prevented from becoming convinced by the conspiracy theory by the implementation of fact-checking. We here assumed that only a small percentage of individuals in the *S* compartment can be convinced per week to reject the conspiracy theory. There are reasonable arguments for this assumption: (i) Not every Twitter user uses Twitter everyday, (ii) fact-checking countermeasures may take some time to have an effect on Twitter users' minds, and (iii) Twitter users may show resistance to fact-checking [46, 47]. Taking these considerations into account, we here tested three levels of $\gamma$ as given by $\gamma_1 = \frac{0.01}{7}$, $\gamma_2 = \frac{0.03}{7}$ and $\gamma_3 = \frac{0.05}{7}$. Accordingly, if considering $\gamma_1 = \frac{0.01}{7}$, approximately one percent of the *S* compartment would progress to the *R* compartment per week.

*Tweet deletion parameter $\zeta$.* The parameter $\zeta$ indicates how many individuals can be removed from the *I* compartment by deleting their tweets. There is evidence that social networks differ in the extent to which they use fact-checking and delete posts [48]. However, it is reasonable to assume that social networks have a fair chance of using (semi-)automatic software to detect suspect tweets. Large-scale deletion of posts is nonetheless difficult, as (i) debate and discussions are essential to social networks, (ii) suspect tweets can be false-positive, and (iii) users may undermine these countermeasures by using different accounts, phrases, hashtags *et cetera*. We therefore assume that social networks are able to perform tweet deletion at low to medium rates. Accordingly, we defined that $\zeta$ should have three levels with $\zeta_1 = \frac{0.06}{7}$, $\zeta_2 = \frac{0.12}{7}$ and $\zeta_3 = \frac{0.25}{7}$.

## Data

We used data collected by Banda et al. [49], who provide aggregated Twitter data beginning from January 1, 2020 ongoing until the time of writing (March 2021). They used the Social Media Mining Toolkit (SMMT; [50]) to constantly listen to the Twitter Stream API to search for specific pandemic-related keywords, e.g. "coronavirus", "covid19", "CoronavirusPandemic" *et cetera*; see [49, for the full list]. Their dataset is considered to be one of the most comprehensive multi-lingual collections of COVID-19 pandemic-related tweets [51],

containing more than 233 million clean tweets in the version used here (version 46; repository: https://zenodo.org/record/4460047). In our study, we used their "hashtags" dataset, which stores the hashtags and their frequencies per day for all clean tweets. Clean tweets mean that retweets are not included. Even though the exclusion of retweets was not optimal for our research purpose, the "hashtags" dataset should nevertheless be a fair proxy for the true dynamics of the "5GCoronavirus" conspiracy theory on Twitter, as it is reasonable to assume that the frequencies of clean tweets are highly correlated with the frequencies of retweets. However, Banda et al. [49] report that they update their dataset every two days, meaning that the dataset should not be much affected by tweet and account deletions. Please note that we have analyzed data on a daily basis from January 1, 2020, until August 15, 2020 ($t_{max}$), which should reflect a fair recording period to study the dynamics of the "5GCoronavirus" conspiracy theory on Twitter. In order to map these dynamics, we identified ten hashtags which (i) clearly indicate that 5G is harmful or connected to SARS-CoV-2, and (ii) were most frequently used. Table 2 shows the selected hashtags and their frequencies. Please note that the term "Incidence/Hashtag incidence", in this context, is defined by the number of times these hashtags were used on a specific day $t_i$.

## Results

### Descriptive analysis

In total, 5611 hashtags were recorded on $n = 202$ days from $t_0$ to $t_{max}$. The first hashtags were observed on January 27, 2020 ($t_0$), which is in line with findings from Bruns et al. [24]. Table 2 provides the frequency of each hashtag. The hashtag incidence time series is depicted in Fig 1A, showing that the "5GCoronavirus" conspiracy theory began at the end of January, followed by an exponential increase with a peak at the beginning of April 2020 and ending approximately at the end of June 2020. The maximal hashtag incidence of 340 was observed on April 8, 2020 ($t_{IMax}$), coinciding almost perfectly with the maximal rolling mean of 256.00 on April 6, 2020 (see Fig 1A). Fig 1A also shows an interesting pattern of reappearing "incidence bursts", meaning that the "5GCoronavirus" conspiracy theory reappeared occasionally after $t_{IMax}$. This observation is in line with the findings from Shin et al. [10], who reported that misinformation tends to come back after initial publication. However, these reappearing "incidence bursts" tend to be much smaller compared to the first peak: the second (beginning of

**Table 2. Frequency table of the ten most used hashtags indicating that 5G is harmful or connected to SARS-CoV-2.**

| Hashtag | Frequency |
| --- | --- |
| #5gcoronavirus | 2171 |
| #5gkills | 1311 |
| #stop5g | 842 |
| #5gcorona | 466 |
| #wuhan5g | 228 |
| #5gvirus | 213 |
| #5gdeadlyweapon | 207 |
| #no5g | 85 |
| #saynoto5g | 43 |
| #5gcononavirus | 45 |
| #(Total) | 5611 |

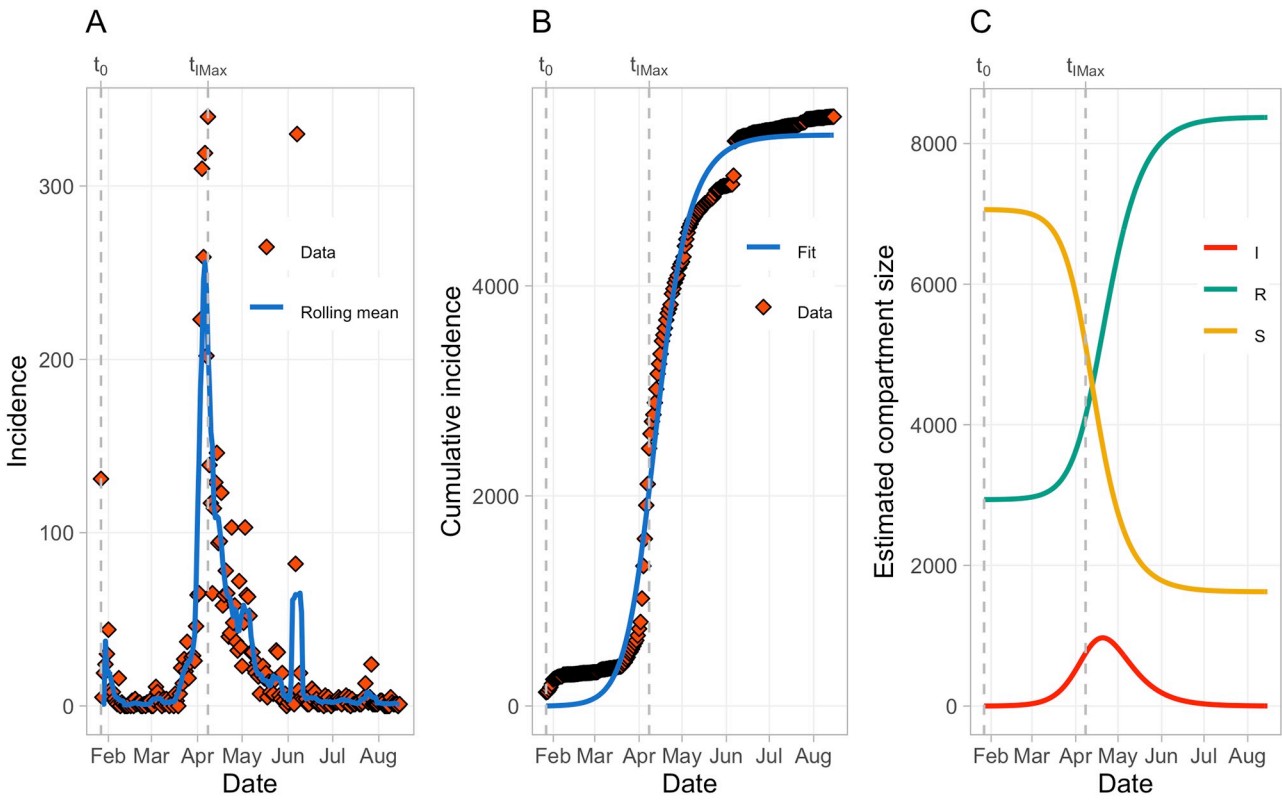

**Fig 1. Acceptable correspondence of observed and predicted hashtag incidence.** (A) Hashtag incidence over time. Please note that "Rolling mean" refers to the simple moving average with the rolling window *k* = 7. (B) Best basic SIR model fit. (C) Model predictions of the compartment sizes over time.

June 2020) and the third (end of July 2020) peaks were accompanied by relatively small maximal deflections in the rolling mean of 65.14 and 7.85, respectively.

## Basic SIR model

Table 3 shows the estimated parameter values yielded by the parameter identification procedure, as well as the error norms $\epsilon_{rel}$ and *MAE*. Fig 1B and $\epsilon_{rel}$ both indicate that the model fit of the basic SIR model is acceptable. The $\epsilon_{rel}$ reported here is in line with results provided by Jin

**Table 3. Parameters and parameter estimations of the best fit basic SIR model.**

| Parameter | Meaning | Estimation |
|---|---|---|
| $\beta$ | infection rate | 0.3 |
| $\alpha$ | recovery rate | 0.11 |
| $\frac{1}{\alpha}$ | infection period | approx. 9 |
| $\mathcal{R}_0 = \frac{\beta}{\alpha}$ | basic reproduction number | 2.7 |
| $S(0)$ | initial susceptible | 7060.94 |
| $I(0)$ | initial infectious | 1.1 |
| $R(0)$ | initial removed | 2936.53 |
| $N = S(0) + I(0) + R(0)$ | population size | approx. 9999 |
| $\epsilon_{rel}$ | relative error in 2-norm | 0.0561 |
| *MAE* | mean absolute error | 194.52 |

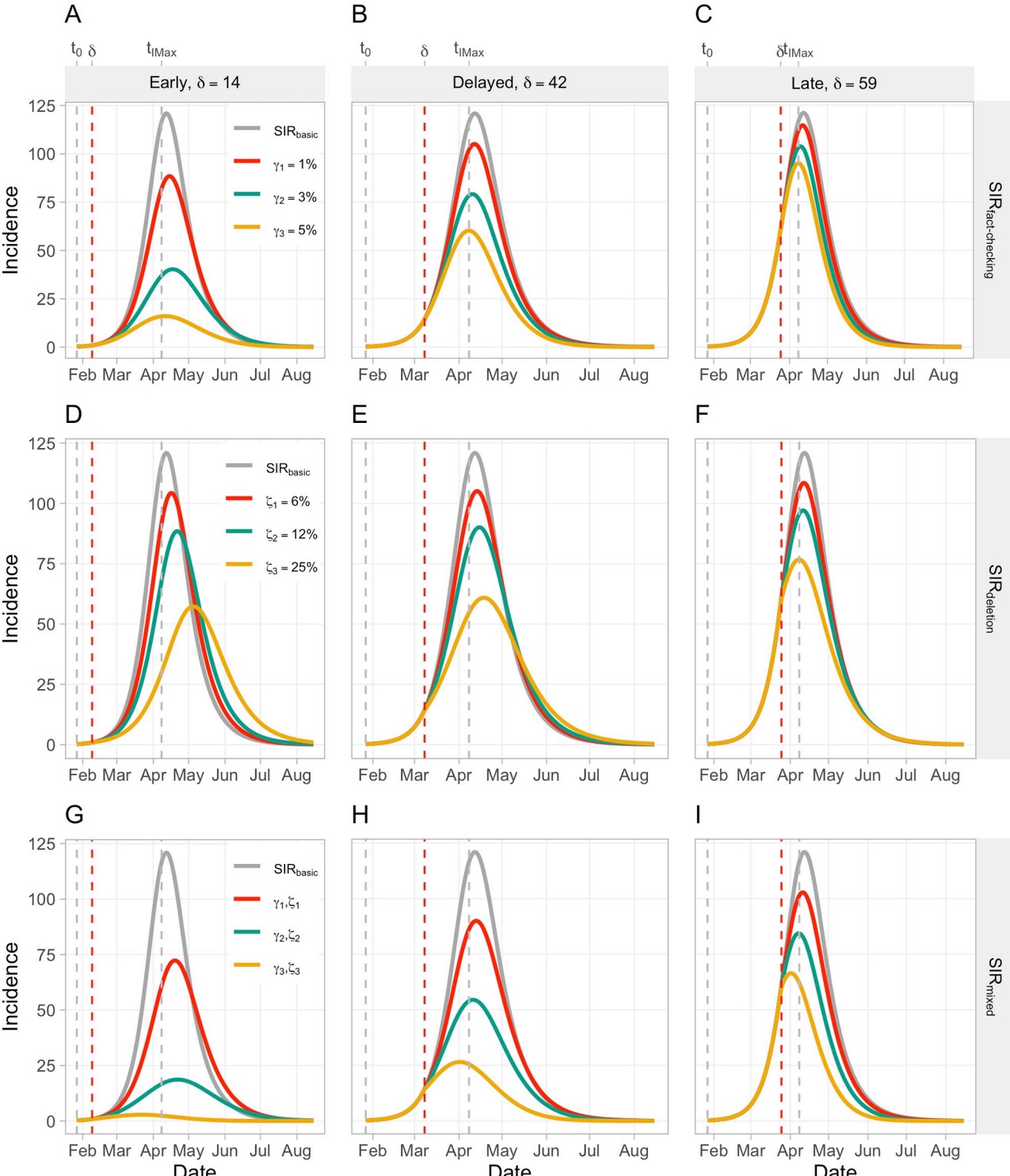

**Fig 2. Predicted incidence over time across different extended SIR models and parameters.** Note that different levels of $\delta$ are depicted column-wise, while different extended SIR models are depicted row-wise. Different levels of $\gamma$ and $\zeta$ vary within each panel A-I. Please also note that, in the interest of simplicity, only corresponding parameter combinations, i.e. $(\gamma_1, \zeta_1)$, $(\gamma_2, \zeta_2)$ and $(\gamma_3, \zeta_3)$, are depicted.

et al. [15], who found similar values of $\epsilon_{rel}$ for their models. We therefore conclude that our basic SIR model shows fair correspondence of observed and predicted incidence, even though it could not fully capture the observed data: As indicated by Fig 1B, the model shows a few systematic divergences from the data due to the (partial) inability of the model to account for the high amplitude of the first peak and the occurrence of reappearing "incidence bursts".

Fig 1C shows the model predictions of the SIR compartment sizes over time. According to these predictions, prevalence is maximal on April 20, 2020 with 971 active cases. This prediction coincides roughly with newspaper reports that the majority of attacks on mobile phone masts were performed in the beginning / middle of April 2020 [22]. The model also shows that there was already a significant proportion of individuals who were recovered/removed at $t_0$, accounting for the fact that not all individuals are necessarily receptive to conspiracy theories. In fact, however, when considering that $S(0) \approx 7060$ and $R(0) \approx 2937$, the model makes the assumption that a majority of individuals (70.62%) was initially susceptible to the "5GCoronavirus" conspiracy theory. This seems unrealistic with respect to prior findings, which indicate that only a minority of individuals is sensitive to conspiracy theories per se [52, for instance]. It is, however, imaginable and underpinned by first evidence (see e.g. [11]) that individuals in social networks aggregate within homogeneously mixed clusters. This leads us to the interpretation that the present model therefore characterizes how the "5GCoronavirus" conspiracy theory propagated through a homogeneous minority of individuals susceptible to conspiracy theories. Finally, this model implies that there was a significant proportion of individuals (23.02%, 1626 individuals) who were susceptible but did not progress to the $I$ compartment, arguably resembling the effects of herd immunity.

## Extended SIR models

We compared the extended SIR models by plotting the incidence curves of all ($SIR_{fact-checking}$ and $SIR_{deletion}$) parameter combinations, and of corresponding parameter combinations for the $SIR_{mixed}$ models. We also show the incidence curve of the basic SIR model as the baseline model. Fig 2 shows the resulting matrix. Please note that the basic SIR model is also referred to as "baseline SIR model" in the following sections.

Furthermore, in order to evaluate the extended SIR models in a more quantitative fashion, we calculated the incidence proportion $IP(t_{max})$, which reflects the probability that a susceptible individual will become infected up to $t_{max}$. Accordingly, this measure characterizes the percentage of susceptible individuals who become infected over the whole time period. We here denote this measure as $IP_f$. Fig 3 shows heatmaps of $IP_f$ given different parameters on the axes. In short, Fig 3 illustrates in a color-coded manner how time-sensitive both countermeasures, fact-checking and tweet-deletion, are. Please note that the $IP_f$ of the baseline SIR model is 0.77, meaning that 77% (5435 individuals) of the susceptible individuals get infected over time in the baseline SIR model. We also present the number of prevented infections, reflecting the absolute reduction of infected individuals in an extended model relative to the baseline SIR model.

**SIR model with fact-checking.** Fig 2A–2C clearly indicates an effect of fact-checking on the incidence, which is more prominent for higher levels of $\gamma$. Accordingly, the more fact-checking is applied, the less individuals become convinced by the conspiracy theory. Importantly, the effect size is strongly $\delta$-dependent, meaning that fact-checking is most effective if applied in early stages (Fig 2A), while late fact-checking is nearly useless, irrespective of the $\gamma$ level (Fig 2C). Considering the effects of an *Early response* combined with a high level of $\gamma$, the incidence curve shows a very moderate progression, indicating that fact-checking alone—if applied early—might have been able to contain the spread of the "5GCoronavirus" conspiracy

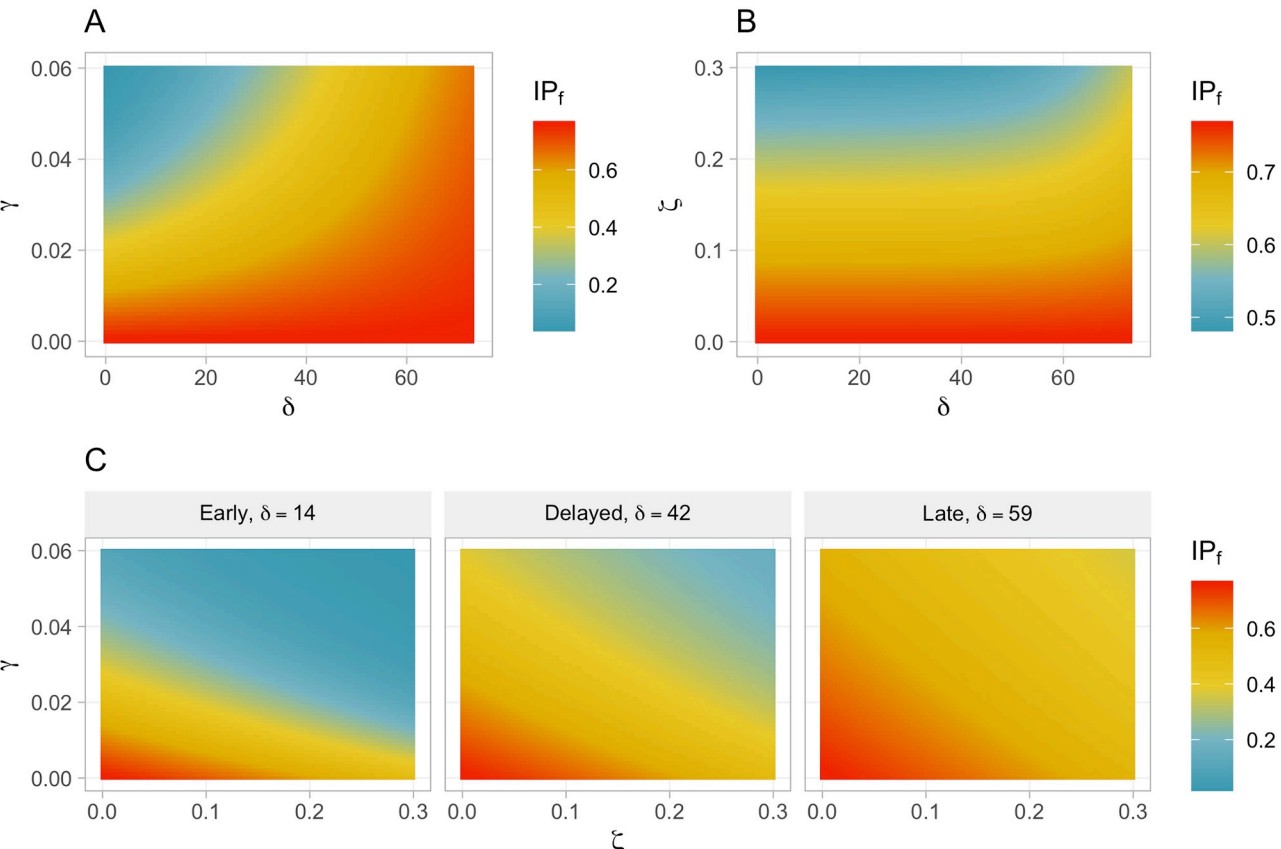

**Fig 3. Heatmaps of $IP_f$ for different parameter combinations.** Please note that fact-checking (panel A) requires early intervention, whereas tweet-deletion (panel B) causes effects which are more stable over time. Please note that panel A and B do not share the same metric, i.e. "blue" in A does *not* correspond to "blue" in B with respect to $IP_f$. Please also note that panel C shows the heatmaps of $\zeta \times \gamma$ given different levels of $\delta$.

theory. This observation is paralleled by the corresponding incidence proportion of $IP_f = 0.16$ and the total number of prevented infections of 4307 (see S2 Table), which provide further quantitative evidence for the superiority of this model over the baseline model. A *Delayed* response, meanwhile, shows significant losses in effectiveness of fact-checking compared to an *Early* response (see Fig 2B), as indicated by the fact that an *Early* but moderate fact-checking response is more effective than a *Delayed* but strong response ($\Delta IP_f = 0.430 - 0.363 = 0.067$).

These results are further supported by Fig 3A, which clearly indicates that fact-checking is effective only if applied early but becomes useless for higher values of $\delta$. More precisely, it is evident from Fig 3A that fact-checking should have been applied within the first 20 days from $t_0$ to keep the $IP_f$ below or close to 0.2. Taking these findings together, we conclude that there is a narrow time window where fact-checking is extremely useful to flatten the incidence curve. Afterwards, the effect of fact-checking alone is substantially attenuated with nearly no effect if applied late.

**SIR model with tweet-deletion.** Fig 2D–2F shows a substantial effect of tweet-deletion on the incidence, which seems to be essentially linearly related to $\zeta$. Interestingly, the effect size is not as $\delta$-dependent as was predicted for the SIR model with fact-checking. Another interesting feature shown in the incidence curves is that there is a slight positive time shift of the peaks for an *Early* response (+ 23 days; see Fig 2D), whereas a slight negative time shift was observed for a *Late* response (−4 days; see Fig 2F). This leads us to the conclusion that tweet-deletion has

potentially a second positive effect in addition to incidence reduction, by prolonging the available response time to conspiracy theories in social networks, providing more time to introduce additional countermeasures. However, it seems that tweet-deletion alone would not have been a sufficient tool to control the spread of the "5GCoronavirus" conspiracy theory: Considering the case of an *Early* response with maximal tweet-deletion, the effect indicated by $IP_f = 0.541$ is relatively weak, compared to the SIR model with fact-checking. The effects of strong tweet-deletion remain relatively stable when considering a *Delayed* response ($IP_f = 0.549$; see Fig 2E), but the effect of the positive peak-shift (+ 6 days) vanishes to a considerable degree.

These results are supported by Fig 3B, showing that the effect of tweet-deletion is relatively stable over time (i.e., fairly robust up to $t_0 + 60$) and declines significantly only in extremely late stages. Accordingly, there is a wider time window in which tweet-deletion can be applied profitably. However, this finding is weakened by the observation that tweet-deletion in general seems to have a limited potential for reducing the incidence, compared to fact-checking.

**SIR model with mixed countermeasures.** Finally, Fig 2G–2I suggests that combining fact-checking and tweet deletion provides an effective tool to reduce the incidence substantially, even if the response is not *Early*. Considering Fig 2H (*Delayed* response), a strong response, as given by $(\gamma_3, \zeta_3)$, can flatten the curve to a moderate level. The superiority of this model over the baseline model can be underpinned quantitatively by considering the corresponding incidence proportion of $IP_f = 0.216$ and the total number of prevented infections of 3911 (see S2 Table). If both countermeasures are applied strongly in an *Early* stage (Fig 2G), the incidence can be reduced to close to zero ($IP_f = 0.029$; absolute only 203 infections), presenting a powerful mechanism which would have been able to contain the spread of the "5GCoronavirus" conspiracy theory. Even an *Early* moderate response, as given by $(\gamma_2, \zeta_2)$, shows satisfying outcomes, as indicated by the incidence curve and the $IP_f$ of 0.204. Nonetheless, even the SIR model with mixed countermeasures fails, as did the previous extended SIR models, to provide evidence that the spread of the "5GCoronavirus" conspiracy theory could have been controlled in a *Late* stage. This is reflected in the incidence curves in Fig 2I. It seems, however, as there is at least some "damage control", as a strong response shows a $IP_f$ of 0.408, which is a significant reduction compared to the baseline SIR model with $IP_f = 0.77$.

Fig 3C further underpins that the effect of countermeasures is highly time-sensitive. While there is a relatively large space of parameter combinations (moderate to strong responses) with good outcomes ("blue" regions) in the left panel (*Early* response), there is only a small "blue" region left—requiring combined strong responses to be reached—when considering a *Delayed* response (panel in the middle). Accordingly, stronger countermeasures are needed if the response is *Delayed* to achieve the same outcome. The right panel of Fig 3C shows that a *Late* response can only inadequately control the incidence. Fig 3C also shows the $\delta$-dependency of both parameters, $\gamma$ and $\zeta$, as $\gamma$ is effective only if applied early, while $\zeta$ is less time-sensitive.

## Discussion

We here demonstrated that the spread of conspiracy theories can be characterized using epidemiological models. This finding is in line with previous research [15, 20], further supporting the potential of epidemiological models, even outside their original scope of infectious diseases (see [53, 54, for instance] and [55, for an overview]). Furthermore, we evaluated the effects that countermeasures could have had on the propagation of the "5GCoronavirus" conspiracy theory on Twitter. Our simulations indicated that (i) fact-checking is an effective mechanism in an early stage of conspiracy theory diffusion, (ii) tweet-deletion shows moderate efficacy and is less time-sensitive than fact-checking, and (iii) combined countermeasures constitute a

promising tool to effectively limit conspiracy theory diffusion. Our simulations also clearly show that response time is critical when dealing with conspiracy theories in social networks. Taking these findings together, we conclude that an early response combined with strong fact-checking and a moderate level of tweet deletion is necessary to control the diffusion of a conspiracy theory through a social network.

The data we have analyzed here potentially implies some limitations with respect to the validity of the underlying epidemiological models. For instance, we used hashtag data instead of data on an individual level. That means that we—strictly speaking—modeled the diffusion of hashtags, but *not* the infection curve of individuals. However, it seems reasonable to assume that hashtag incidence is a fair proxy of the "true" incidence curve. Previous research suggests that hashtags can be modeled by epidemiological models too [20]. Acquiring data on an individual level nevertheless seems reasonable, as epidemiological models axiomatically deal with individuals. In the present study, we initially conducted a large-scale download of tweets, requesting tweet content using the Twitter API from more than 140 million tweet ids. A subsequent interim analysis of the downloaded tweets revealed that the downloaded data was substantially affected by data dropouts (60% missing data), shaping our decision to base our analyses on the "hashtag" dataset. Future studies may overcome these issues by downloading the tweets with a smaller time lag or even in parallel to the evolution of the conspiracy theory in question. At the same time, it is also important to note that false positive tweets may play a significant role too: An individual who tweeted a suspect hashtag ("5GCoronvarirus", for instance) need *not* necessarily be convinced by the conspiracy theory, but could use the hashtag in a humoristic fashion, or could state that there is no evidence for the conspiracy theory. Our interim analysis of the downloaded tweets also supports the presence of false positive tweets, indicating that future studies should make efforts to "filter out" these tweets by applying text classification. Text classification of tweets has been intensively studied with respect to sentiment analysis [56], showing that sentiment analysis is very challenging due to the extreme heterogeneity of the tweets and limited length (max. 280 characters) of the tweets. There is also research about the detection of misinforming tweets [57, 58], indicating the potential of machine learning to estimate tweet credibility. Associated with the problem of false positive tweets are social bots, which evidently facilitate the spread of misinformation through social networks [59] but do not constitute human entities. However, the application of machine learning and respective tools for both the detection of misinforming tweets [60, for instance], and bots [61] seems reasonable and may help to better understand the "true" diffusion of misinformation through social networks.

There is also a need for further theoretical and empirical work in order to produce more valid models of epidemiological conspiracy theory diffusion in social networks. Our simple SIR model showed acceptable model fit, but it failed to account for both the extreme peak at the beginning of April 2020 and the reappearing "incidence bursts". More complex models may provide solutions to the weaknesses of the simple SIR model. Furthermore, taking into account the heterogeneity of contact patterns within social networks may also help to better understand the "true" spread of conspiracy theories in social networks. Relaxing the homogeneous-mixing assumption of the basic SIR model by using network-based approaches may help to account for this issue [62]. However, even this simple model was generally able to characterize the propagation of the "5GCoronavirus" conspiracy theory. Future studies should focus on (i) improving the underlying epidemiological model, (ii) advancing the parameter identification procedure [63, for an infomrative article about this issue], and (iii) increasing the number of studied conspiracy theories. Implementing these points may help to learn more about the underlying processes of when and how conspiracy theories propagate through social networks.

Taking into account recent advances in memory and human factors research [64, for instance] may also help to improve the underlying epidemiological model: Ognyanova et al. [65] found that fake news exposure was associated with a decline in mainstream media trust. Transferring this finding to the epidemiological model would imply that there exists a significant proportion of susceptible individuals who are unresponsive to fact-checking at all. There is also evidence that if an individual has once established a false belief, "curing" it can be extremely challenging [66], meaning that—within an epidemiological framework—infected individuals should not be seen as recovered after the infection period, but potentially sensitive to become 'infectious' again. In this context, it is interesting to note that individuals who believe in one conspiracy theory are also more likely to believe other, unrelated rumors [67]. Implementing such empirical findings into the model may prove challenging, but would increase their ability to adequately reflect real-world relationships.

More generally, the extended SIR models used in this study should be seen as starting points with the need of further theoretical elaboration and empirical work. With respect to the fact-checking model, the degree to which the model captures real-world relationships currently remains to be explored. For instance, the values of the parameter $\gamma$ were defined arbitrarily, as it is unclear how effective fact-checking actually is. Furthermore, it is unlikely that fact-checking affects susceptible individuals only: Conceivably, fact-checking could also convince an infected individual to reject the conspiracy theory in question or motivate removed individuals to convince susceptible and infected individuals that the conspiracy theory in question is wrong. One more important point is that the analyzed Twitter data potentially already includes the effects of fact-checking and tweet deletion: Twitter evidently deleted tweets which called for a "breakdown" of 5G towers [29]. While our results suggest that implementing such measures may be effective in countering the spread of conspiracy theories, the loss of data through deletion poses a problem for gaining a clear picture of individuals' usage behaviour of the corresponding hashtags. Projects such as "FakeNewsNet" [68] or "FacebookHoax" [33], which provide large-scale datasets of fake news posts before the introduction of countermeasures, can potentially address this problem, especially as the recorded fake news posts are extremely heterogeneous with respect to the underlying fake news story.

The present findings, that fact-checking and tweet-deletion seem to differ in their efficiency as countermeasures against misinformation, should be interpreted against psychological models of belief formation and human memory—especially those that consider psychological mechanisms for the formation and persistence of false memories or false knowledge [69, 70]. For instance, the relative inefficacy of fact-checking at later stages of spread might indicate that to be effective, fact-checking needs to take place before much memory consolidation of misinformation has occurred. Our findings are also potentially relevant for the increasingly urgent question of how to combat deliberate misinformation attacks [64] that are solely intended to support the goals of the attacker, and that are increasingly implemented via automated bots.

Arguably, misinformation attacks unfold their deceptive efficiency via basic psychological mechanisms. For instance, in a digital world with many competing pieces of information, more salient bits of information grab more attention, which could explain in part why false stories are much more likely to be retweeted on Twitter than true stories [71]. Early (mis)information also contributes to establishing an "anchor"—a mental model against which subsequent incoming information is interpreted; once a belief is established, a confirmation bias may influence the belief holder to actively seek information that confirms the belief, and to discount information that is inconsistent with the belief (for a model of cognitive mechanisms involved in processing misinformation attacks, see [64]). As misinformation attacks are

becoming more sophisticated and dangerous, the design and scientific evaluation of counter-measures, though currently in its infancy, is becoming a high-priority challenge for society. Immediate and repeated communication of true facts is probably more efficient than the repeated correction of wrong information, but far more research is needed to translate the emerging cognitive psychology of information attacks into an efficient and versatile set of defensive measures [64, 72]. Increasing evidence on the negative social consequences of conspiracy theories, e.g., for adherence to pandemic restrictions and vaccination recommendations, for the intention to engage in politics or for reducing one´s carbon footprint [73], underlines that efforts towards science-based solutions for addressing misinformation attacks may not only be well-invested, but ultimately vital.

## Conclusion

The present study has provided important evidence for the effects that countermeasures can have on the propagation of conspiracy theories through social networks. We found that fact-checking is an extremely powerful tool in the early stages of conspiracy theory diffusion, but fails in later stages. Tweet-deletion is less powerful, but also less time-dependent than fact-checking. We therefore conclude that fact-checking is the better choice if the conspiracy theory is not in an exponential increase yet. Tweet-deletion should be applied strongly if the conspiracy theory is already propagating rapidly through the social network, and moderately otherwise. In view of the difficulty of identifying novel conspiracy theories, a combined implementation of multiple counter-measures is most likely to prove successful.

## Supporting information

**S1 Table. Guesses for the parameters of the basic SIR model to be optimized over.** (PDF)

**S2 Table. Outcomes of the tested extended SIR models.** (PDF)

## Acknowledgments

We thank Thoraf Kauk for providing additional computational resources. We also thank Christine Nussbaum and Frank Nussbaum for their comments on the methods section of the manuscript.

## Author Contributions

**Conceptualization:** Julian Kauk, Helene Kreysa.

**Data curation:** Julian Kauk.

**Methodology:** Julian Kauk, Stefan R. Schweinberger.

**Project administration:** Helene Kreysa.

**Resources:** Stefan R. Schweinberger.

**Software:** Julian Kauk.

**Supervision:** Helene Kreysa, Stefan R. Schweinberger.

**Validation:** Stefan R. Schweinberger.

**Visualization:** Julian Kauk.

**Writing – original draft:** Julian Kauk, Helene Kreysa.

**Writing – review & editing:** Helene Kreysa, Stefan R. Schweinberger.

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
