## [Decision Letter · Decision Letter 0]

20 May 2021

PONE-D-21-11717

Understanding and countering the spread of conspiracy theories in social networks: Evidence from epidemiological models of Twitter data

PLOS ONE

Dear Dr. Kauk,

Thank you for submitting your manuscript to PLOS ONE. After careful consideration, we feel that it has merit but does not fully meet PLOS ONE’s publication criteria as it currently stands. Therefore, we invite you to submit a revised version of the manuscript that addresses the points raised during the review process.

Both reviewers greatly appreciated the work and found merit in it. At the same time, they highlighted areas where the manuscript could be improved. These mainly revolve around providing additional details, explanations, and discussion on the adopted methods. Please, do take the thorough reviewers' comments in great consideration when preparing your revision.

We look forward to receiving your revised manuscript.

Kind regards,

Stefano Cresci

Academic Editor

PLOS ONE

Journal Requirements:

2. Please upload a copy of Supporting Information Table 1 which you refer to in your text on page 10.

Reviewers' comments:

Reviewer's Responses to Questions

**Comments to the Author**

1. Is the manuscript technically sound, and do the data support the conclusions?

Reviewer #1: Yes

Reviewer #2: Yes

2. Has the statistical analysis been performed appropriately and rigorously? 

Reviewer #1: Yes

Reviewer #2: Yes

3. Have the authors made all data underlying the findings in their manuscript fully available?

Reviewer #1: Yes

Reviewer #2: Yes

4. Is the manuscript presented in an intelligible fashion and written in standard English?

Reviewer #1: Yes

Reviewer #2: Yes

5. Review Comments to the Author

Reviewer #1: The present manuscript is technically sound, and the study is overall well-presented and conducted. Statistical analyses have been performed appropriately and rigorously and all the data underlying the findings in the manuscript are publicly available, as well as the software employed, making the study fully reproducible. The manuscript is presented in an intelligible fashion and written in standard English. In this paper the authors model the diffusion of a conspiracy theory on Twitter using the SIR epidemic model. They also develop three extensions of the model to include two possible countermeasures and a combination of the two. The models are fitted to Twitter hashtags data during a period of almost nine months, considered to be the one regarded by the birth and death of this conspiracy theory. Results clearly suggest that epidemic modelling of conspiracy theories diffusion on online platforms may be a good choice for a better understanding of the phenomena and for the simulation of possible solutions. The hypothesis and results are well put inside of the existing literature. There are still some limitations in the development of such models, since some hypothesis on how agents enter in the different compartments are very strong and the supporting evidence is poor, and in the collection and analysis of the real data, which fail to capture a lot of important aspects due to their simplicity. All of these concerns are however highlighted by the authors themselves in the discussion on the limitations and possible extensions of the present work, which may be considered a simple starting point for more refined analyses.

The main contribution of this work is to extend the literature on epidemic modelling of information diffusion with a specific concern on conspiracy theories and therefore fake news, which are claimed to have some specific characteristics with respect to true fact diffusion. One baseline model - the SIR model - is considered and three extended models are developed and described. All the models are fitted to real data with fair results.

I do not believe there are any major issues to take care of to improve the present work as it is presented.

I believe there are some minor issues that need to be reconsidered to improve the present work. In general, I think that some aspects need to be explained in greater detail. For example, it is not clear how the countermeasure here considered would work in a real setting and how they would produce the specific result claimed (bringing an individual from one compartment to another). Moreover, in the Parameter Identification paragraph, I think the concept of Incidence should be defined and contextualized better as well as the variable $n$ and also it is understandable that the time unit is a single day since the data includes the frequency of each hashtag per day, but never explicitly stated. In my opinion also the Data section should be revised. All the information seems to be present, but the discourse is a bit unclear: I do not know if it is necessary to explain why in the end the choice went on the hashtag data set instead of full tweets, but maybe I would focus more on the description of the data and how despite limitations they may be enough to understand the underlying phenomena. I also think that in the description of the Results some improvements may come from a more detailed description of the values from the real data to the values estimated by the models, especially when considering plots and tables, which are not thoroughly described within the text or the caption. Maybe some of the key evidence from the tables/figures in terms of numerical values may be added to the text and not left to the reader to understand. For example, in figure 3A some of the percentages of IPf may be reported and compared across models and to the baseline value, which I also think should be stated at the beginning of this section and not after the description of all three extensions, to give the reader a chance to understand the differences from the extended models without having to go back. For what concerns the Discussion section, I think general conclusions and limitations are well assessed, however I think some points should have been presented also in the results section and not here for the first time, for example things such as the models failing to capture the second smaller peak, how smaller is the peak with respect to the first one...

I have no additional remarks to make.

Reviewer #2: I think this is a very interesting paper that deserves publication in PONE after few minor issues will be addressed.

1) Reference [1] is not so enough to support the sentence after which it is mentioned. I think something like Zarocostas, John. "How to fight an infodemic." The lancet 395.10225 (2020): 676. would be better.

2) The authors do not specificy what they mean by incidence and cumulative incidence (I_{cum}). I guess it is the total outreach of posts/hashtags.

3) The authors should provide further details regarding the estimation of the SIR parameters (especially N whose inference seems to be crucial for the results of the model)

4) Do the authors expect different results by relaxing the assumption of full mixing of the SIR model? Social networks are generally sparse while (if I am not wrong) the SIR in the version discussed in the paper doesn't consider this aspect. (see for instance The echo chamber effect on social media M Cinelli, GDF Morales, A Galeazzi, W Quattrociocchi, M Starnini Proceedings of the National Academy of Sciences 118 (9))

5) The correct reference for Ref 6 is: The COVID-19 Social Media Infodemic [...] Scientific Reports volume 10, Article number: 16598 (2020)

6. PLOS authors have the option to publish the peer review history of their article (what does this mean?). If published, this will include your full peer review and any attached files.

Reviewer #1: No

Reviewer #2: No

---

## [Author Response · Author response to Decision Letter 0]

16 Jun 2021

Dear Reviewers of PLOS ONE,

Please find our response to your points in "Response to reviewers.pdf". We gratefully thank you for your comments and suggestions which much helped to improve the manuscript.

On behalf of all authors,

Sincerely,

Julian Kauk

---

## [Decision Letter · Decision Letter 1]

2 Aug 2021

Understanding and countering the spread of conspiracy theories in social networks: Evidence from epidemiological models of Twitter data

PONE-D-21-11717R1

Dear Dr. Kauk,

We’re pleased to inform you that your manuscript has been judged scientifically suitable for publication and will be formally accepted for publication once it meets all outstanding technical requirements.

Kind regards,

Stefano Cresci

Academic Editor

PLOS ONE

Reviewers' comments:

Reviewer's Responses to Questions

**Comments to the Author**

1. If the authors have adequately addressed your comments raised in a previous round of review and you feel that this manuscript is now acceptable for publication, you may indicate that here to bypass the “Comments to the Author” section, enter your conflict of interest statement in the “Confidential to Editor” section, and submit your "Accept" recommendation.

Reviewer #1: All comments have been addressed

Reviewer #2: All comments have been addressed

2. Is the manuscript technically sound, and do the data support the conclusions?

Reviewer #1: Yes

Reviewer #2: Yes

3. Has the statistical analysis been performed appropriately and rigorously? 

Reviewer #1: Yes

Reviewer #2: Yes

4. Have the authors made all data underlying the findings in their manuscript fully available?

Reviewer #1: Yes

Reviewer #2: Yes

5. Is the manuscript presented in an intelligible fashion and written in standard English?

Reviewer #1: Yes

Reviewer #2: Yes

6. Review Comments to the Author

Reviewer #1: I believe all the points assessed by the Editor and the Reviewers have been properly considered and taken into account by the authors. Additional explanations and information were added where needed, i.e. in the description of specific countermeasures already used by social networking platforms or in the explanation of the process of parameter estimation.

I also believe the Data and Results sections have been considerably improved and give a more concise and clear insight into the work described in the present manuscript. I think moving the explanation on why hashtag data were used instead of individual level data to the Discussion section improved readability and enriched the Discussion section valuably.

Reviewer #2: (No Response)

7. PLOS authors have the option to publish the peer review history of their article (what does this mean?). If published, this will include your full peer review and any attached files.

Reviewer #1: No

Reviewer #2: No

---

## [Editor Report · Acceptance letter]

4 Aug 2021

PONE-D-21-11717R1 

Understanding and countering the spread of conspiracy theories in social networks: Evidence from epidemiological models of Twitter data 

Dear Dr. Kauk:

I'm pleased to inform you that your manuscript has been deemed suitable for publication in PLOS ONE. Congratulations! Your manuscript is now with our production department. 

Kind regards, 

on behalf of

Dr. Stefano Cresci 

Academic Editor

PLOS ONE